# Prognosis and Immunological Characteristics of PGK1 in Lung Adenocarcinoma: A Systematic Analysis

**DOI:** 10.3390/cancers14215228

**Published:** 2022-10-25

**Authors:** Yuechao Yang, Huanhuan Cui, Deheng Li, Yang Gao, Lei Chen, Changshuai Zhou, Mingtao Feng, Wenjing Tu, Sen Li, Xin Chen, Bin Hao, Liangdong Li, Yiqun Cao

**Affiliations:** 1Department of Neurosurgery, Fudan University Shanghai Cancer Center, Shanghai 200032, China; 2Department of Oncology, Shanghai Medical College, Fudan University, Shanghai 200032, China

**Keywords:** PGK1, lung adenocarcinoma, tumor microenvironment, biomarker, immunotherapy

## Abstract

**Simple Summary:**

Immunotherapy has become a major treatment for lung adenocarcinoma. Better understanding of the tumor microenvironment (TME) in lung adenocarcinoma is needed in order to better treat it with this type of therapy. In this study, we evaluate the associations of PGK1 with prognosis and immunological characteristics in lung adenocarcinoma using bioinformatic analysis methods. The results of this study provide clues for a potential immunometabolic combination therapy strategy that may improve the immunotherapeutic efficacy of LUAD.

**Abstract:**

**Background**: Aerobic glycolysis plays a key role in tumor metabolic reprogramming to reshape the immune microenvironment. The phosphoglycerate kinase 1 (PGK1) gene codes a glycolytic enzyme that converts 1,3-diphosphoglycerate to 3-phosphoglycerate. However, in lung adenocarcinoma (LUAD), the role of PGK1 in altering the tumor microenvironment (TME) has not yet been determined. **Methods**: Raw data, including bulk DNA and mRNA-seq data, methylation modification data, single-cell RNA-seq data, proteomics data, clinical case characteristics survival, immunotherapy data, and so on, were obtained from multiple independent public data sets. These data were reanalyzed to uncover the prognosis and immunological characteristics of PGK1 in LUAD. **Results**: We found that PGK1 mRNA and protein were considerably over-expressed in LUAD compared to normal tissue and that high PGK1 expression is associated with poorer prognostic outcomes in LUAD. The enrichment analysis of PGK1 co-expressed genes in lung adenocarcinoma revealed that PGK1 may be involved in hypoxia, metabolism, DNA synthesis, cell cycle, PI3K/AKT, and various immune and inflammatory signaling pathways. Furthermore, PGK1 is also linked to the recruitment of numerous immune cells, including aDC (dendritic cells), macrophages, and neutrophils. More importantly, PGK1 was highly expressed in immunosuppressive cells, including M2 macrophages, Tregs, and exhausted T cells, among others. Finally, higher PGK1 expression indicated significant correlations to immune checkpoints, TMB (tumor mutation burden), and high response to immunotherapy. **Conclusions**: The presented findings imply that PGK1, as a glycolysis core gene, may be important for the modification of the immune microenvironment by interacting with the tumor metabolism. The results of this study provide clues for a potential immunometabolic combination therapy strategy in LUAD, for which more experimental and clinical translational research is needed.

## 1. Introduction

Lung adenocarcinoma is the leading cause of cancer-related death worldwide as the most common histological sub-type, accounting for about 40% of lung cancer incidences [1,2]. Despite advances in the treatment of lung adenocarcinoma, the vast majority of lung cancer patients eventually relapse. The main reason for treatment failure is that lung adenocarcinoma can develop a high metastatic potential, which is responsible for approximately 90% of patient deaths from lung cancer [3]. To provide highly accurate predictions of patient survival and the response to individualized therapy, new biomarkers and therapeutic targets for the improved detection of and new targeted treatments for LUAD are urgently needed.

PGK1 is a serine (Ser)/threonine (Thr) kinase, which catalyzes the formation of ATP in the aerobic glycolysis pathway. The combined control of PGK1 on glycolysis, mitochondrial metabolism, and autophagy drive malignant tumor progression, according to previous studies [4]. Furthermore, by modulating ATP and 3-PG levels, PGK1 has a rate-limiting function in coordinating energy generation, biosynthesis, and redox equilibrium [5]. As a gene activated by increased oxidative stress and vascular damage, PGK1 has been linked to a wide range of diseases, including Parkinsonism, hereditary non-spherocytic hemolytic anemia, neurological impairment, myopathy, and malignant tumors [6,7,8,9].

A significant number of studies have shown that genes participating in both glucose and glutamine transport and metabolism, including PGK1, PFK1, and so on, exert tremendous influences on the metabolic adaptation, proliferation, and metastasis of cancer cells [10,11,12]. The high proliferation of tumor cells determines their high metabolic rate, which leads to competition with immune cells for nutrients in the tumor microenvironment, thus mediating immune cell dysfunction and immune tolerance [13]. To execute progression and metastasis, cancer cells go through metabolic evolution to maximize nutrient utilization for bioenergetic and biosynthetic demands, survive harsh tumor microenvironments (TMEs), and escape immunosurveillance [14]. PGK1 has also been shown to play an important role in immune regulation in several tumors [15,16], but a comprehensive systematic analysis of PGK1 in remodeling the LUAD immune microenvironment, particularly by the use of large-scale, multi-omics, and single cell level data, has not been reported.

In this work, through the systematic analysis of PGK1 in lung adenocarcinoma, we identify that the over-expression of PGK1 may affect the immune microenvironment of LUAD through a tumor metabolic immune interaction, suggesting a potential strategy to improve the efficacy of other immunotherapies; namely, by inhibiting PGK1 or aerobic glycolysis to modify the tumor immune microenvironment. This study may provide some clues for metabolic immunotherapy strategies.

## 2. Materials and Methods

### 2.1. Utilizing Open-Access Resources to Analyze PGK1 mRNA and Protein Expression

The mRNA expression of PGK1 was investigated in a pan-cancer context using TCGA (The Cancer Genome Atlas) database in TIMER2.0 (http://timer.cistrome.org/, accessed on 10 March 2022) [17]. In the meanwhile, three further verified data sets from the GEO database (GSE10245, GSE32863, and GSE7670) were downloaded (https://www.ncbi.nlm.nih.gov/geo/, accessed on 12 March 2022). Furthermore, the HPA database (http://www.proteinatlas.org, accessed on 14 March 2022) was used to examine protein expression levels. HPA045385 (Sigma-Aldrich, St. Louis, MI, USA) and CAB010065 (Sigma-Aldrich) were among the antibodies utilized in the HPA database (Santa Cruz Biotechnology). The UALCAN (http://ualcan.path.uab.edu/, accessed on 20 March 2022) tool was used to analyze the total protein and phosphorylated protein expression of PGK1 in LUAD, for which the data were obtained from the Clinical Proteomic Tumor Analysis Consortium (CPTAC) [18].

### 2.2. Clinical Features and Prognosis of PGK1 Using Public Databases

The LUAD data were provided by TCGA. This information included RNA-seq results, as well as clinical characteristics (including both survival data and phenotype). The R software was utilized to investigate the link between PGK1 and LUAD clinical characteristics. The ggplot2 package in R was used for visualization. GEPIA2.0 (http://gepia2.cancer-pku.cn/#index, accessed on 4 April 2022), a platform for TCGA data visualization, was also utilized to evaluate the effect of PGK1 on OS in a pan-cancer context and to create Kaplan–Meier survival curves. Meanwhile, the relationship between PGK1 expression and survival in LUAD was also investigated and validated using the Kaplan–Meier plotter (http://kmplot.com/analysis/, accessed on 8 April 2022), incorporating LUAD from the TCGA and GEO datasets (GSE14814, GSE19188, GSE29013, GSE30219, GSE31210, GSE3141, GSE31908, GSE37745, GSE50081) and the PrognoScan database (GSE3141, GSE13213, GSE11117, GSE31210; http://dna00.bio.kyutech.ac.jp/PrognoScan/index.html, accessed on 9 April 2022) [19], which offers gene expression survival statistics across a substantial number of publicly accessible cancer microarray data sets. The threshold was adjusted to a Cox p-value of <0.05. ROC curves were analyzed using the pROC package in R.

### 2.3. Co-Expression Module Identification and Pathway Analysis 

Initially, the stat package in R was used to screen for PGK1 co-expressed genes and the clusterProfiler and org.Hs.eg.db packages were used to perform GO function enrichment analysis of the top 100 co-expressed genes connected with PGK1 expression in the TCGA database. FunRich conducted the KEGG pathway analysis [20,21].

### 2.4. Protein–Protein Interaction Network Analysis 

STRING (https://cn.string-db.org/, accessed on 15 April 2022) was used to build the network of PGK1 and its probable co-expressed genes in LUAD from TCGA and Cytoscape-MCODE was used to display the protein–protein interaction (PPI) network [22]. Cytoscape-cytoHubba was also used to identify core genes and establish the PPI network. Furthermore, Gene Set Cancer Analysis (GSCA; http://bioinfo.life.hust.edu.cn/web/GSCALite/, accessed on 18 April 2022) was used to analyze the expression, mutation, methylation, and pathway enrichment of the top 10 genes in LUAD [23]. 

### 2.5. Genetic Alteration Analysis 

The genetic modification status of PGK1 was investigated using cBioportal database analysis (http://www.cbioportal.org, accessed on 22 April 2022), containing the TCGA data [24]. The difference in PGK1 methylation between LUAD and para-cancer tissues was investigated using the DNMIVD database (http://119.3.41.228/dnmivd/index/, accessed on 27 April 2022) [25]. Furthermore, the “SCNA” component of TIMER analysis revealed the relationship between PGK1 immune cell infiltration level alteration and gene copy number [26]. 

### 2.6. Immune Infiltration Analysis

We used TISIDB (http://cis.hku.hk/TISIDB/index.php, accessed on 2 May 2022) to analyze the correlation between PGK1 and immunomodulators, chemokines, and lymphocytes, which is an online integrated repository portal collecting abundant human cancer datasets from the TCGA database [27]. In addition, SSGSEA in R was employed to quantify the correlation between PGK1 expression and immune infiltration in the LUAD [28]. As a validation database, TIMER (https://cistrome.shinyapps.io/timer/, accessed on 8 May 2022) was used to evaluate the impact of immune infiltration on the clinical prognosis in LUAD [26]. Furthermore, we also used GEPIA2021 (http://gepia2021.cancer-pku.cn/sub-expression.html, accessed on 15 May 2022) to investigate the effect of PGK1 expression on different sub-types of immune cells [29].

### 2.7. PGK1 Expression Analysis in Non-Small Cell Lung Cancer at Single-Cell Level 

The multi-dimensional features of NSCLC infiltrating lymphocytes were discovered using the Peking University database (http://lung.cancer-pku.cn/index.php, accessed on 18 May 2022), which includes deep single cell transcriptome data as well as comprehensive T cell receptor information [30]. A pooling method was used for normalization, as implemented in the R function computeSumFactors. The data were in log_2_ space after they had been normalized. There were 12,598 genes and 7183 cells in the final expression table. The expression of PGK1 was indicated using box plots for tissue type, cell type, tissue type, cell type, and T cell cluster; the T cell cluster was additionally reduced using the R package Rtsne. GSE131907 from scTIME Portal was used to perform single-cell sequencing analysis of the tumor immune microenvironment to evaluate PGK1 expression in tumor cells and immune cells in LUAD. (http://sctime.sklehabc.com/unicellular/search, accessed on 25 May 2022) 

### 2.8. PGK1 Expression in Relation to TMB and MSI (Microsatellite Instability) Analysis 

cBioPortal (https://www.cbioportal.org/, accessed on 2 June 2022) database analysis was used to analyze the correlation between PGK1 expression and TMB/MSI [24]. TIDE (tumor immune dysfunction and exclusion; http://tide.dfci.harvard.edu/, accessed on 8 June 2022) was used to explore the common immune-predictive score of PGK1 in NSCLC, which is based on tumor expression profiles before treatment, the TIDE module can estimate a variety of published transcriptome biomarkers to predict patient response [31,32].

## 3. Results

### 3.1. PGK1 Is Highly Expressed in the Vast Majority of Malignant Tumors

RNA-sequencing data from TCGA data were used to analyze the mRNA expression profile of PGK1 in a pan-cancer context. When comparing the expression of PGK1 normal samples in the TCGA and GTEx (Genotype-Tissue Expression) databases with comparable tumor samples in the TCGA database, PGK1 was found to be considerably over-expressed in a variety of cancers (Figure 1A), including LUAD. Meanwhile, in a few types of cancer, including KICH, *PGK1* mRNA expression was low in tumor tissues.

### 3.2. Both mRNA and Protein of PGK1 Were Highly Expressed in LUAD

As in the training database mentioned above, compared to normal tissues, LUAD presented considerably higher levels of PGK1 mRNA expression (Figure 1B). To further verify the higher expression of PGK1 in LUAD, additional validated data from GEO was employed, including two data sets (GSE7670 and GSE32863) comprising a total of 182 samples (91 tumor samples and 91 control normal tissues). The results demonstrated that the mRNA expression level of PGK1 in LUAD was considerably greater than that in normal tissues after normalizing the expression profile (Figure 1C). In addition, the mRNA expressions of PGK1 in LUAD and LUSC were explored utilizing GSE10245. Interestingly, there were no obvious differences in the expression of PGK1 between LUAD and LUSC. Besides, the protein expression of PGK1 was explored in lung cancer and pan-cancer contexts. A total of 21 patients received CAB010065 and HPA045385 antibodies in lung cancer. The IHC intensity of these patients (n = 21) were as follows: IHC staining was seen in six patients with high intensity, one with moderate intensity, and eight with low intensity. The staining of PGK1 with a CAB010065 antibody in lung cancer is shown in Figure 1D, showing high/medium staining in 7 of 11 lung cancer patients. The staining of PGK1 in generalized carcinoma is shown in Appendix A, indicating that PGK1 is expressed to a certain extent in most cancer species and is significantly expressed in lung cancer. Furthermore, the UALCAN tool was used to analyze the total protein and phosphorylated protein (S203) expression of PGK1 in tumor and normal tissues of LUAD. Various literature confirms that phosphorylation of PGK1 at S203 is associated with the occurrence and development of tumors and poor prognosis, including breast cancer and glioma [33]. ERK-dependent PGK1 S203 phosphorylation and subsequent PIN1-mediated cis-trans isomerization mediated the EGFR/K-Ras/B-Raf induced mitochondrial translocation of PGK1 [34]. The results indicated that both phosphorylated and total PGK1 protein were increased in tumor tissues and the expression level increased with the progress of tumor grade and stage. Interestingly, both the phosphorylated and total protein expression of PGK1 was highest in LUAD patients aged 21–40 years, whereas the expression level showed a downward trend with an increase in age (Figure 1E,F). Because the above data sources were all from bulk RNA-seq or whole tissue microarray, in order to reduce the influence of non-tumor cells in the tumor microenvironment, we analyzed single-cell sequencing data from GSE131907 in Figure 1G,H to verify the expression difference of PGK1 between tumor cells and normal epithelial cells from the single-cell level (Appendix A). The results suggested that the expression of PGK1 in tumor cells is significantly higher than that in normal epithelium. At the same time, the expression of PGK1 in brain metastatic and lymph node metastatic tumor cells is higher than that in primary tumor and normal epithelium cells, which may be the reason for the increased expression of PGK1 with the increase in the tumor grade and stage. Especially, the Venn map of the differential genes of tumor cells in different metastatic tumor cells vs. lung primary tumor cells indicated that PGK1 was a metastatic specific gene (Figure 1H). 

### 3.3. High PGK1 Expression Predicts Poor Prognosis in Pan-Cancer and LUAD

To identify the prognosis value of *PGK1* expression, we used GEPIA2.0 to analyze the relationship between the expression of PGK1 and pan-cancer overall survival (OS; Figure 2A). The results indicated that PGK1 expression was associated with poor prognosis in a variety of cancers, including LUAD.

According to the TCGA data, there is a link between PGK1 over-expression and poorer overall survival in LUAD patients (HR = 1.4, *p* = 0.021; Figure 2B). In addition, survival data from the Kaplan–Meier plotter database were analyzed. The over-expression of PGK1 (probe ID 227068_at) was consistently linked to a poorer prognosis in patients with LUAD, in terms of OS, according to the findings (HR = 1.67, 1.3–2.14; *p* = 4.3 × 10^−5^; Figure 2C). Then, four cohorts (GSE3141, GSE13213, GSE11117, GSE31210) of patients were obtained from the GEO database in order to validate the poor prognostic value of PGK1 expression in LUAD (Figure 2D). These findings demonstrated that PGK1 over-expression was linked to a worse outcome in LUAD patients. It was proposed that the following hypothesis should be tested: in patients with LUAD, PGK1 might be used as a biomarker to assess the LUAD prognosis.

We investigated the association between PGK1 expression and the clinical traits of LUAD patients in the TCGA data sets, in order to learn more about the significance and underlying processes of PGK1 expression in cancer. Table 1 demonstrates the important clinical features of PGK1 determined in the TCGA study. It can be seen that, with tumor progression (including TNM stage and tumor grade), the expression of PGK1 increased. In addition, the expression of PGK1 differed with the number of smoking years in tumor patients. Further, we visualized the expression of PGK1 in some clinical features of lung adenocarcinoma using the ggplot2 package in R (Appendix A). In order to explore the predictive effect of PGK1 expression on the outcome of LUAD, the pROC package in R was used for analysis. The predictive ability of the variable PGK1 presented a certain accuracy (AUC = 0.813, CI: 0.771–0.855; Appendix A).

### 3.4. PGK1 Co-Expressed Genes in LUAD Are Enriched in Tumor Development, Metabolism, and Immune-Related Pathways

In order to determine the biological processes related to PGK1 in LUAD, GO/KEGG enrichment analysis was performed on PGK1 and its co-expressed genes in 594 LUAD patients from TCGA data, for which the stat package in R package was utilized to screen the co-expressed genes of PGK1, satisfying a total of 2484 co-expressed genes with cor > 0.3 and *p* < 0.05. As a result of the screening, 878 genes were shown to be substantially positively associated with the expression of PGK1, whereas 1606 genes were discovered to be significantly negatively linked with the expression of PGK1 (*p* < 0.05). PGAM1, which encodes a mutase that catalyzes the reversible conversion of 3-phosphoglycerate (3-PGA) to 2-phosphoglycerate (2-PG1) in the glycolytic pathway, demonstrated the greatest relationship when the screening threshold was cor > 0.7 and *p* < 0.05 (cor = 0.700, *p* = 5.027 × 10^−80^). GO function enrichment analysis of the top 100 co-expressed genes associated with PGK1 expression was carried out using ClusterProfiler and org.Hs.eg.db. Under the settings of *p* < 0.05 and q < 0.05, PGK1 co-expressed genes were involved in 180 biological processes (BP), 32 cell components (CC), and 8 molecular functions (MF). The top ten messages for BP, CC, and MF are shown in the histogram (Figure 3A). PGK1 co-expressed genes were mostly involved in the hypoxia process, metabolism, immune microenvironment, and tumor microenvironment (TME; Figure 3A). KEGG pathway analysis was performed using FunRich, which demonstrated that the co-expression of PGK1 was primarily associated with cell cycle, metabolism, anaerobic process, and other pathways. Notably, KEGG pathway enrichment results showed that PGK1 and its co-expressed genes were enriched in immune-related pathways, including C-MYC pathway, Vpu-mediated degradation of CD4, Antigen Processing-Cross presentation, Class I MHC-mediated antigen processing and presentation, and adaptive immune system (Figure 3B).

### 3.5. PPI Enrichment Analysis in Co-Expressed Genes of PGK1

STRING was used to create a network of PGK1 and its possible co-expressed genes in LUAD and Cytoscape-MCODE was used to exhibit a PPI network containing 92 nodes and 261 edges (Figure 3C). Furthermore, we used Cytoscape-cytoHubba to determine the 30 genes with the most nodes, which formed the core position in the network, and established a PPI network containing 30 nodes and 172 edges (Figure 3D). As can be seen from the results, most genes communicating closely with PGK1 participate in biological processes such as metabolism and cell proliferation, including ENO1, HIF1A, PKM2, and LAHA. Furthermore, GSCA was used to analyze the expression, single nucleotide mutation, copy number mutation, methylation, and pathway enrichment of the top 10 genes in LUAD. These genes were significantly over-expressed in LUAD, including PGK1 (Figure 4A). We utilized the pathway activity module of GSCA to compare gene expressions between pathway active (activated) and repressed (suppressed) groups, as defined by pathway scores. Interestingly, these genes mainly activated EMT, cell cycle, and apoptosis, while inhibiting RAS/MAPK and PI3K/AKT in 10 well-known tumor-related pathways (Figure 4B). The heat map showed what percentage of these genes activated or inhibited cancer-related pathways, where pathway A (red) represents the percentage of cancers in which the pathway may activated by given genes, while inhibition is shown, in a similar way, by pathway I (blue). The results indicated that these genes were more likely to activate the cell cycle pathway (Figure 4C). Pathway scores revealed particular pathway activation and inhibition percentages (number of activated or inhibited cancer types/32 × 100%) in the global proportion of malignancies, in which PGK1 had an influence on the pathway among 32 types of cancer (Figure 4D) among the genes affecting these pathways across various cancers. Appendix A shows the single nucleotide mutation, copy number variation, and methylation situations of these genes. The above results suggested that PGK1 and its related functional genes have a synergistic effect in the occurrence and development of tumors. 

### 3.6. Genetic Alteration Analysis of PGK1 across Pan-Cancer and LUAD

The genetic modification status of PGK1 was identified in several tumor samples from the TCGA cohorts. As shown in Appendix A, individuals with uterine corpus endometrial carcinoma with “mutation” as the major type had the greatest PGK1 alteration frequency (>5%). Interestingly, all instances of kidney chromophobes with genetic abnormalities (2% frequency) presented PGK1 copy number deletion. There was an approximately 2% mutation frequency of PGK1 in LUAD, where the “mutation” type of genomic alteration accounted for most of them. Detailed information regarding PGK1 mutation is further presented in Appendix A. We found that missense and splice mutations were the main types of PGK1 genetic alteration in LUAD. The X214 splice site is shown in the 3D structure of the PGK1 protein in Appendix A. In addition, we used cBioportal to analyze the effects of gene mutations and copy number mutations on mRNA expression of PGK1. The mRNA expression of PGK1 was analyzed at pan-cancer and LUAD levels, respectively, and the results showed that the mRNA expression of PGK1 of “Not profiled” was higher than that of “No mutation” at the pan-cancer level. The results of copy number variation indicated that the mRNA expression of “Gain” was higher than that of “Shallow Deletion” and “Diploid” (Appendix A). In LUAD, there was no significant difference in RNA expression between different mutation types and non-mutation. At the same time, the “Gain” of the copy number variation was higher than that of the “Shallow Deletion” (Appendix A). These results indicate that mutations in the PGK1 gene may not cause changes at the mRNA level and the differential expression of PGK1 in lung adenocarcinoma may be caused by other regulatory mechanisms, such as copy number variation, epigenetic modification, genomic transcription, or protein translation, etc. DNMIVD was used to analyze the difference in PGK1 methylation between tumor and para-cancer tissues in LUAD, and the results indicated that there was no significant difference between the two groups (Appendix A), suggesting that the differential expression of PGK1 may not be caused by promoter methylation. In TIMER, SCNA module analysis showed that several kinds of immune cell infiltration levels seemed to associate with PGK1 gene copy number alterations in LUAD, including macrophages, CD4+ T cells, B cells, and neutrophils (Appendix A).

### 3.7. PGK1 Expression Reshapes LUAD Immune Microenvironment

Tumor-infiltrating lymphocytes have been shown to be an independent predictor of cancer survival and sentinel lymph node status [35]. Tumors reshape the immune microenvironment to promote immune escape through metabolic–immune interactions with immune cells [36]. Based on the GO/KEGG enrichment analysis results, depicted in Figure 4, PGK1 co-expressed genes were mainly involved in the anaerobic (hypoxia) process, metabolism, and immune-related pathways. We used TISIDB to assess the relations between the abundance of tumor-infiltrating lymphocytes (TILs) and expression of PGK1 in order to examine which kinds of TILs might be regulated by PGK1. Meanwhile, TISDB was used to assess the connections between PGK1 expression and immunomodulators/chemokines/lymphocytes. We found that PGK1 was significantly positive correlated with immunological infiltration. We visualized the most correlated immune cells, immune checkpoints, cytokines, chemokines, and receptors (Figure 5A–F). For immunostimulators: THFRSF9 (rho = 0.313, *p* = 4.46 × 10^−13^) and THFRSF13B (rho = −0.281, *p* = 9.27 × 10^−11^); for immunoinhibitors: CD274 (rho = 0.31, *p* = 2.2 × 10^−13^) and CD160 (rho = −0.249, *p* = 1.08 × 10^−8^); for major histocompatibility complex (MHC): B2M (rho = 0.25, *p* = 9.41 × 10^−9^) and TAPBP (rho = −0.15, *p* = 0.000655); for Lymphocytes: Act DC (rho = 0.412, *p* < 2.2 × 10^−16^) and Eosinophil (rho = −0.313, *p* = 4.29 × 10^−13^); for Chemokines: CCL7 (rho = 0.44, *p* < 2.2 × 10^−16^) and CCL14 (rho = −0.35, *p* = 3.04 × 10^−16^); and, for Chemokine receptors: CCR1 (rho = 0.214, *p* = 9.16 × 10^−7^) and CXCR5 (rho = −0.213, *p* = 1.13 × 10^−6^). Furthermore, to verify the results of TISIDB, ssGSEA in R was used to re-evaluate the relationship between PGK1 expression and the abundance of immune cell infiltration. PGK1 was shown to be related to Th2 cells, active dendritic cells (aDCs), macrophages, and neutrophils (Figure 6A). The strongest connection was seen between PGK1 and Th2 cells, which play an important role in B cell-mediated humoral immunity (Spearman r = 0.513) (Figure 6B,C). This indicates that the high expression of PGK1 in bulk tumor tissues was positively correlated with the infiltration of Th2 and active dendritic cells in the tumor microenvironment. To increase the reliability of the data, we used the TIMER database for secondary verification. The “Gene” component analysis revealed that, whereas PGK1 expression had no discernible relationship with tumor purity, it was strongly linked to infiltrating levels of macrophages, neutrophils, and dendritic cells in LUAD (Figure 6D). We also evaluated the effect of immune infiltration on the clinical prognosis of LUAD patients. High amounts of dendritic cells, B cells, and neutrophils were related to a better prognosis in LUAD patients, according to our findings (Figure 6E). Then, we used the GEPIA2021 database to investigate the effect of PGK1 expression on different sub-types of the same immune cell type in tumor and normal tissues and found that PGK1 was highly expressed in immunosuppressive and resting cells, including M2 macrophages, resting dicentric cells, resting mast cells, and so on (Appendix A). Interestingly, comparing the PGK1 expression level between tumor and normal tissues according to different immune cell subtypes, the results showed that, with regard to the M1 macrophages and the activated NK cells—known as immunoactived cells—PGK1 presented a higher expression in the tumor than normal tissues, which may be related to more intricate regulatory mechanisms that might exist (Appendix A). These results indicated that, by influencing the number of tumor-infiltrating immune cells, PGK1 may influence LUAD and its clinical outcomes, the specific molecular regulation mechanisms related to which need to be confirmed through further experimental studies.

### 3.8. Validation of PGK1 Expression in Exhausted and Immunosuppressive T Cells in Non-Small Cell Lung Cancer in Single-Cell Sequencing Data

Based on the role of PGK1 in remolding the immune microenvironment, as indicated above, we further used the Peking University database (http://lung.cancer-pku.cn/index.php, accessed on 18 May 2022)—which includes deep single cell transcriptome data as well as complete T cell receptor relevant data, portraying the multi-dimensional characteristics of NSCLC-infiltrating lymphocytes—in order to evaluate PGK1 expression in global T cells of non-small cell lung cancer at the single-cell level. When t-SNE was applied to the expression data, it revealed that PGK1 was strongly expressed in the majority of CD4+ and CD8+ T cell clusters (Figure 7A–D). The results indicated that PGK1 was more highly expressed in the tumor-infiltrating tissue type than normal and peripheral blood tissue types. The greatest PGK1 expression levels were seen in CD8-C6-LAYN and CD4-C7-CXCL13 clusters, suggesting fatigued T cells, whereas CD8-C4-GZMK and CD4-C5-EOMES showed the lowest expression, both of which are pre-exhausted T cells. In addition, the expression of PGK1 was higher in CD4-C9-CTLA4, suggesting that the high expression of PGK1 may be related to the activation of Treg. Furthermore, the expression of PGK1 with tissue type, cell type, and tissue type × cell type are showed as box plots in Figure 7C,E,F. In four different types of T cells, including CD4+CD25HI, CD4+CD25INT, CD4+CD25LOW, and CD8+ T cells, the expression of PGK1 was higher in CD4+CD25HI group. The results above indicated that the high expression of PGK1 shapes an immunosuppressive microenvironment. Interestingly, the expression of PGK1 in Tregs in normal tissues was higher than that in malignant tissues, which needs further investigation (Appendix A). The TIDE database was used to analyze the effect of PGK1 expression in tumor tissues on the CTL number prediction ability of patient survival with immune therapy. The level of CTL predicted the poor prognosis of lung adenocarcinoma under the condition of higher PGK1 expression, while the level of CTL has no effect on the prognosis of lung adenocarcinoma in the case of lower PGK1 expression (Figure 8A). At the same time, we also performed a single cell sequencing analysis of the tumor immune microenvironment in lung adenocarcinoma in GSE131907 in a scTIME Portal and evaluated the expression of PGK1 in tumor cells and immune cells. The results showed that PGK1 was highly expressed in both tumor cells and immune cells. The expression of PGK1 in immunosuppressive T cells were higher, such as in CD4-CTLA4-Treg and CD8-PDCD1, whereas the expression of PGK1 was lower in other immune effector cells. Additionally, we visualized that the infiltration level of effector T cells, including CD8-GZMK-FOX, CD8-ZNF683, and CD8-GZMK, were negatively correlated with PGK1 expression in tumor cells (Figure 8B–D), which also verified our idea that a higher expression of PGK1 in tumor cells could promote the generation of an immunosuppressive microenvironment.

### 3.9. PGK1 Expression Is Related to TMB and Predicts the Effect of Immunotherapy 

MSI and TMB are common immune predictive scores, which can predict the therapeutic effect of immunotherapy. Therefore, we used cBioportal to evaluate the MSI and TMB scores of PGK1 in LUAD. The results demonstrated that the TMB score was higher in the PGK1 genomic alteration group than in the non-alteration group, while the MSI score showed the opposite result (Figure 9A). Then, TIDE was used to explore the performance of PGK1 in some common immune prediction scores. Two studies met the criteria after screening. The results showed that PGK1 performed well in most of the immune prediction scores in LUAD, especially in terms of the TIDE, CD8, IFNG, and Merck18 scores (Figure 9B); further, the AUC scores of the two studies all exceeded the random line, suggesting a good prediction effect. The data of PGK1 altered in LUAD were obtained from the the cBioportal database, including 29 patients in the altered group and 537 patients in the unchanged group. The altered group accounted for 5% of all LUAD patients and mRNA high was the most common type of change (Figure 9C).

## 4. Discussion

A malignant tumor, to some extent, is thought to be a metabolic disorder [37]; therefore, it is critical to comprehend the roles of metabolic-related genes in tumors. Growing evidence has suggested that the metabolic interaction between tumor and microenvironment plays a key role in tumor development and immune escape [38]. The metabolic function of PGK1 is mainly involved in glycolysis and various metabolic events. PGK1 dysfunction can cause metabolic reprogramming and the Warburg effect [39]. In addition to cell metabolic control, PGK1 participates in a variety of biological processes, including DNA repair, autophagy, and angiogenesis, which leads to the complex role of PGK1 in promoting the occurrence and development of tumors [40]. In addition, it has also been reported that the functions of PGK1 inside and outside cells are different, and the high expression of PGK1 in cells leads to the proliferation of tumor cells. However, the high extracellular expression of PGK1 inhibits malignant tumors by inhibiting angiogenesis. PGK1 is also associated with chemoradiotherapy resistance and poor prognosis in cancer patients.

The results of the present study revealed that PGK1 expression is significantly higher in patients with LUAD, in comparison to normal tissues, and a high expression level of PGK1 was associated with the poor prognosis of patients with LUAD. Through co-expression analysis, we determined that most of the co-expressed genes of PGK1 were glycolysis-induced genes, including GAPDH, ENO1, PKM, and so on, as well as hypoxia-induced genes, such as HIF1A. GO/KEGG pathway enrichment and PPI analysis showed that the co-expressed genes of PGK1 were mainly enriched in hypoxia, cell cycle, metabolism, and immune environment-related pathways. Similar to the results of the co-expression analysis, the PPI results showed that the most closely communicating genes with PGK1 are involved in biological processes such as cell proliferation and metabolism, including ENO1, HIF1A, PKM2, and LDHA, suggesting that PGK1 plays an important role in the growth, metastasis, and metabolism of LUAD [41,42,43,44]. Interestingly, we found that these genes can inhibit the TSC/mTOR, RAS/MAPK, and PI3K/AKT pathways using the GSCA database, which are frequently activated in human cancers and control the uptake and utilization of a variety of nutrients, including glucose, amino acids, nucleotides, and lipids [45], while the aerobic glycolysis process—also known as the Warburg effect—is generally considered to be activated as a downstream effect, as regulated by the PI3K signaling pathway [46]. 

There may be several reasons for this seemingly contradictory phenomenon. First, there exists a strong heterogeneity among different tumor types, even within the same tumor specimen [47,48]. Meanwhile, the tumor microenvironment also has great heterogeneity. Different tumor types and separated spatial positions within the same tumor tissue may have heterogeneous (or even completely opposite) metabolic patterns and molecular regulatory mechanisms [36,49,50]. Second, tumor bulk-sequencing data failed to distinguish cells at the single-cell level among tumor and microenvironment. There may exist competition for nutrient absorption, determining the heterogeneous metabolism and molecular regulatory networks among these cells [51,52]. Last, but not least, the RAS/MAPK/PI3K/AKT/mTOR pathway can also be regulated by a variety of upstream and downstream regulators, including hypoxia, receptor tyrosine kinases, integrins, reactive oxygen species (ROS), and various transcription factors, which regulate the gene transcription of the key points in the pathway, constructing a complex and extensive molecular regulatory network [53]. Therefore, we can speculate that the inhibition of these pathways may be due to some negative feedback mechanisms caused by the over-expression of these glycolytic genes in tumor cells. Ultimately, this interesting phenomenon and specific regulatory mechanism need to be confirmed in the laboratory. Moreover, we also found that the co-expressed genes of PGK1 may also be enriched in immune-related pathways. However, there was no significant difference in the methylation level, suggesting that the high expression of PGK1 may not be caused by methylation in LUAD. 

Furthermore, the TISIDB, TIMER, GEPIA2021, R, and other analysis results suggested that PGK1 was strongly correlated with immune-associated molecules and immune cell infiltration. TISIDB was used to analyze the correlation between PGK1 and immunomodulators, of which TNFRSF9—a receptor contributing to the clonal expansion, survival, and development of T cells—was most strongly co-expressed with PGK1. Recent research has supported the finding that, as a known activation marker for antigen-specific Tregs, TNFRSF9^+^ cells also form a major part of the functional tumor Tregs [30]. These results suggest that the high expression of PGK1 may be related to the infiltration and activation of Tregs, contributing to the formation of an immunosuppressive tumor microenvironment. As the strongest co-expressed immune-inhibitor with PGK1, CD274 encodes PD-L1 protein expression, one of the most effective immune checkpoint inhibitor targets. A recent study has indicated that the combination of a TNFRSF9 agonist and PD-L1 (CD274) can effectively activate and amplify tumor-specific cytotoxic T cells to enhance tumor control and killing, suggesting that PGK1 may serve as a potential immunotherapy target or can enhance the anti-tumor effect of PD-L1 [54]. As for chemokines, PGK1 presented the strongest correlation with CCL7, a secreted chemokine which attracts macrophages during inflammation and NSCLC metastasis, suggesting the potential mechanism of the correlation between PGK1 expression and macrophages [55]. With regard to chemokine receptors, PGK1 had the strongest correlation with CCR1, the ligands of which include myeloid progenitor inhibitory factor-1 (MPIF-1), monocyte chemoattractant protein 3 (MCP-3), macrophage inflammatory protein 1 alpha (MIP-1 alpha), and regulated on activation normal T expressed and secreted protein (RANTES), implying that it plays an important role in recruiting immune cells to tumor sites. Meanwhile, ssGSEA in the GSVA package of R was used to analyze the relationship between PGK1 and immune cell infiltration levels in LUAD, the results of which indicated that PGK1 cells were strongly related to Th2, DC, macrophages, and so on. Furthermore, we utilized GEPIA2021 to explore the expression of PGK1 in the different polarization sub-types of these immune cells and found that PGK1 expression was significantly higher in macrophage M2 and lower in the activated dendritic and mast cells in LUAD, suggesting that PGK1 might mediate the immune escape of lung adenocarcinoma tumor cells, as a result of the interaction between highly proliferative tumor cells and immune microenvironment cells in competing for nutrients (e.g., glucose); however, the specific molecular mechanism requires further experimental elucidation [14,56]. NSCLC single-cell sequencing data further demonstrated that PGK1 was expressed higher in exhausted T and T regulatory cells. These results suggest that the poor prognosis of LUAD with high PGK1 expression may be related to the effect of PGK1 on the immune infiltration of different types of T cells [30]. Meanwhile, we also observed that the expression of PGK1 was higher in CD4-C9-CTLA4 than CD4-C8-FOXP3, suggesting that the high expression of PGK1 may be related to the activation of Treg. Interestingly, the expression of PGK1 in Treg in normal tissues was higher than in malignant tissues, according to our findings. This might be due to the metabolic flexibility of Treg. Glycolysis, FAO, or oxidative phosphorylation have been found to be increased in Treg in recent research [57]. Additionally, some research demonstrated that the immunosuppressive function of Treg depends on lipid metabolism, but its proliferation and migration depend on glycolysis. Interestingly, enhanced glycolysis, FAO, or oxidative phosphorylation may increase Treg differentiation and proliferation, but the inhibition of glycolysis has little effect on Treg, implying that the metabolic deprivation of Treg by the tumor causes metabolic remodeling, but not enough to inhibit Treg from functioning in the tumor microenvironment [57]. This might explain why PGK1 expression in Treg is lower in normal tissues, but higher than that in other T cells. In contrast, the expression of PGK1 in Teff cells (CD-C3-GNLY, CD8-C3-CX3CR1) was low, suggesting that the glycolysis of Teff cells might be inhibited by the scarce available carbohydrates in TME, the potential reason being that the proliferating tumor cells consume a large amount of nutrients in the microenvironment, resulting in a lack of sufficient nutritional substrates for immune cells to activate the glycolysis process to proliferate and kill tumor cells [58]. These results suggest that the poor prognosis of PGK1 expression in LUAD may be related to the effect of PGK1-mediated immune escape, for which further laboratory confirmation is required. Some research has demonstrated that enhanced tumor glycolysis is linked to tumor immune treatment resistance and that inhibiting glycolysis can boost the anticancer immunological action of T cells [13]. This exactly corresponds to our immune-predictive score results, where PGK1—a critical gene of glycolysis—had a positive effect on LUAD immunotherapy. 

## 5. Conclusions

In this study, we found that PGK1 has a negative effect on the survival of lung adenocarcinoma patients, which might be due to PGK1’s mediating interaction between tumor metabolism and immunoediting, suggesting that PGK1 could serve as a potential immune-combination therapy target. 

## Figures and Tables

**Figure 1 cancers-14-05228-f001:**
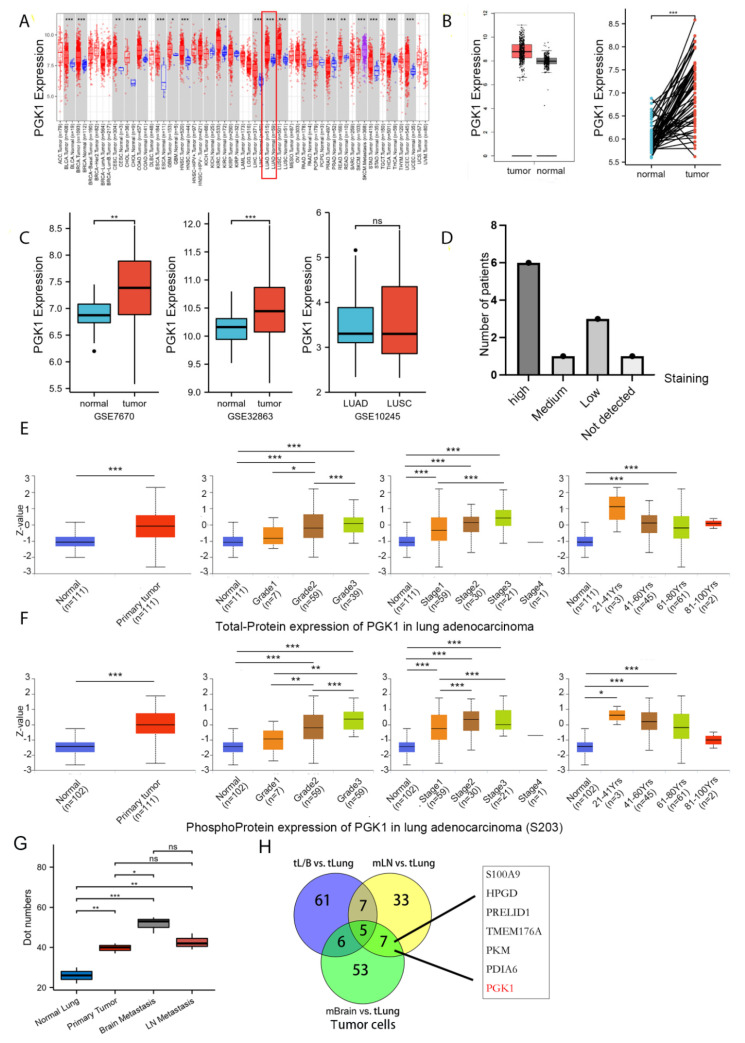
PGK1 expression levels in pan-cancer and LUAD contexts: (**A**) From the TCGA database, PGK1 mRNA expression levels in pan-cancer context. (**B**) PGK1 mRNA expression levels in LUAD, in comparison to healthy tissue taken from the TCGA database. (**C**) Levels of PGK1 mRNA expression in LUAD from two different GEO data sets, as well as in LUAD and LUSC. (**D**) Expression levels of PGK1 protein in LUAD and LUSC from the HPA database. (**E**) Expression levels of PGK1 total-protein in different clinical characteristics of LUAD from the CPTAC database. (**F**) Levels of PGK1 phosphoprotein expression in various LUAD clinical traits taken from the CPTAC database. (**G**) The expression of PGK1 in different tissues including primary and metastatic lesions was obtained from the single-cell database GSE131907. (**H**) The Venn map of differential gene of tumor cells in different metastatic tumor cells vs. lung primary tumor cells. tL/B vs. tLung: Lung metastasis vs. primary lung cancer; mBrain vs. tLung: Brain metastasis vs. primary lung cancer; mLN vs. tLung:Lymph node metastasis vs. primary lung cancer. (dot numbers are the number of cells in a fixed area) (* *p* < 0.05, ** *p* < 0.01, *** *p* < 0.001, ns > 0.05).

**Figure 2 cancers-14-05228-f002:**
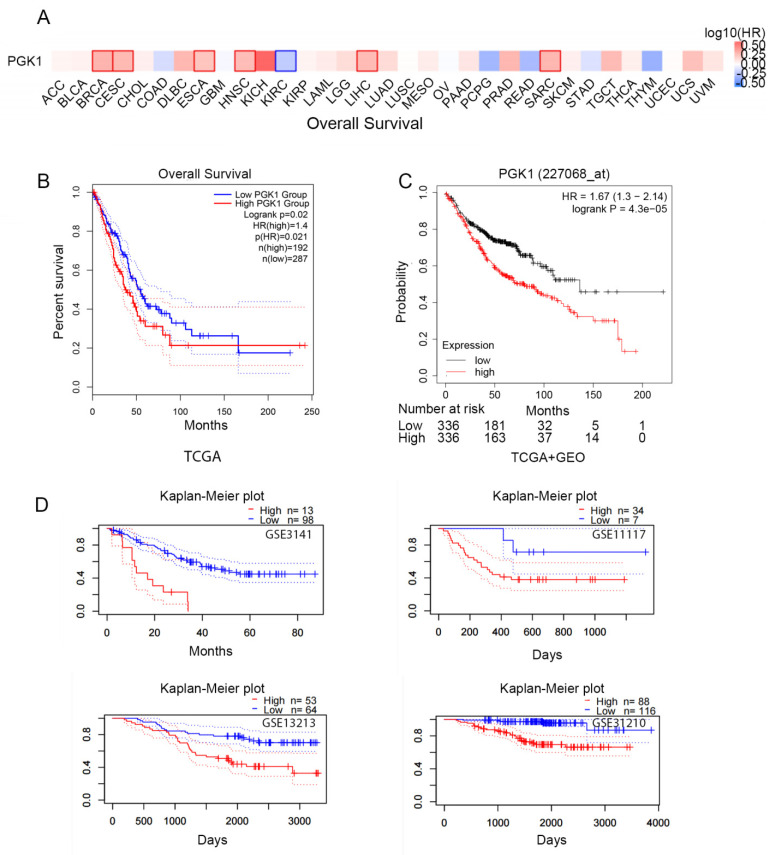
Prognostic value of the mRNA expression of PGK1. (**A**) Relationship between the expression of PGK1 and pan-cancer overall survival (OS) from TCGA database. (**B**) Prognostic value of the mRNA expression of PGK1 in LUAD from TCGA database (GEPIA2.0). (**C**) The Kaplan–Meier curve of PGK1 LUAD obtained from integrated data of TCGA and GEO (Kaplan–Meier plotter). (**D**) The poor prognostic value of PGK1 expression in LUAD obtained from the GEO database (PrognoScan).

**Figure 3 cancers-14-05228-f003:**
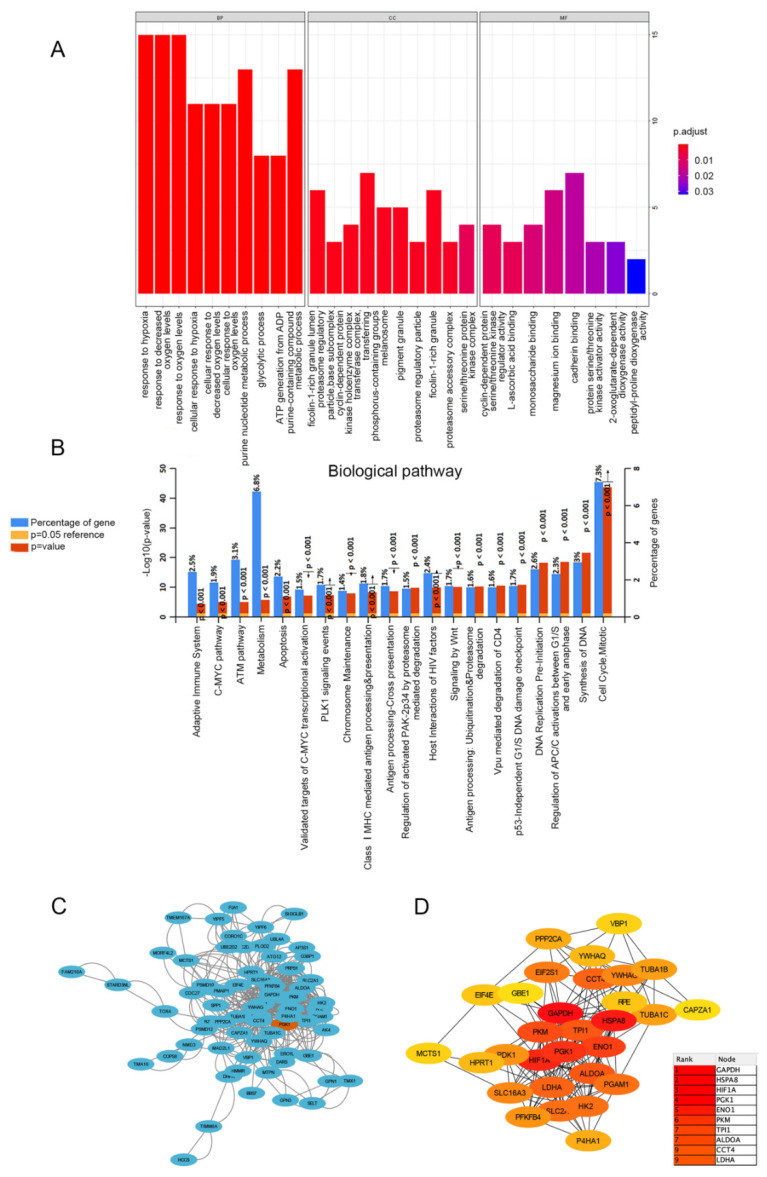
Functionally annotated PGK1 co-expression module. Top 100 GO annotation (**A**) and KEGG pathway terms (**B**). Cytoscape was used to build the PPI network of co-expressed genes (**C**). The most important module was acquired from the PPI network, which included 30 nodes and 172 edges (**D**).

**Figure 4 cancers-14-05228-f004:**
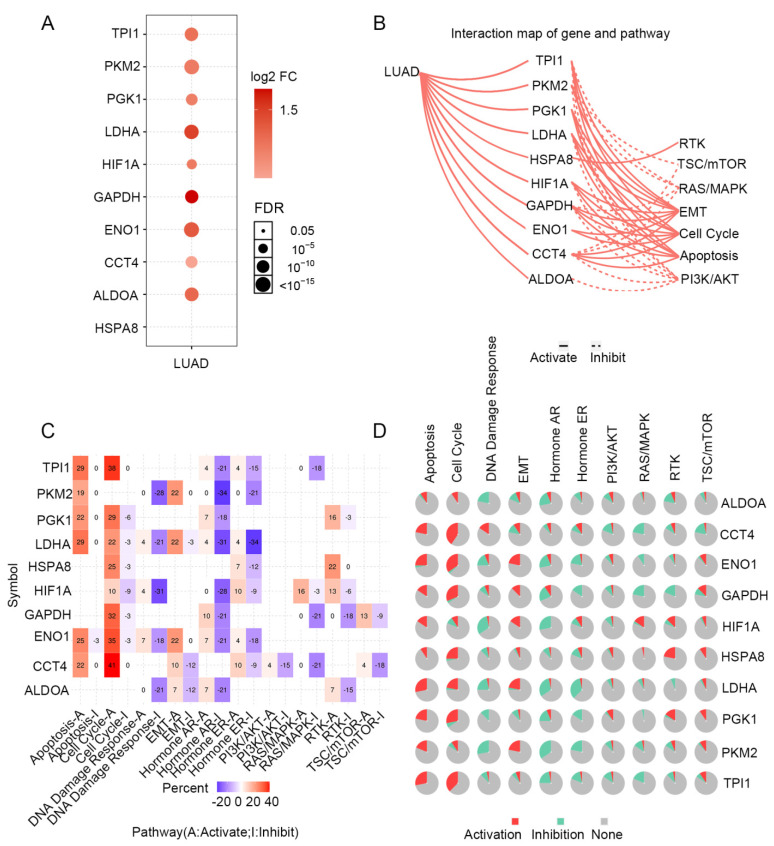
Expression mutation and functional annotation analysis of top 10 co-expressed genes of PGK1. (**A**) mRNA expression of co-expressed genes in LUAD. (**B**) Interaction map of co-expressed genes and pathways. (**C**) The percentage of these genes activating or inhibiting a cancer-related pathway. (**D**) The percentage of cancers in which a gene has an effect on the pathway among 32 cancer types.

**Figure 5 cancers-14-05228-f005:**
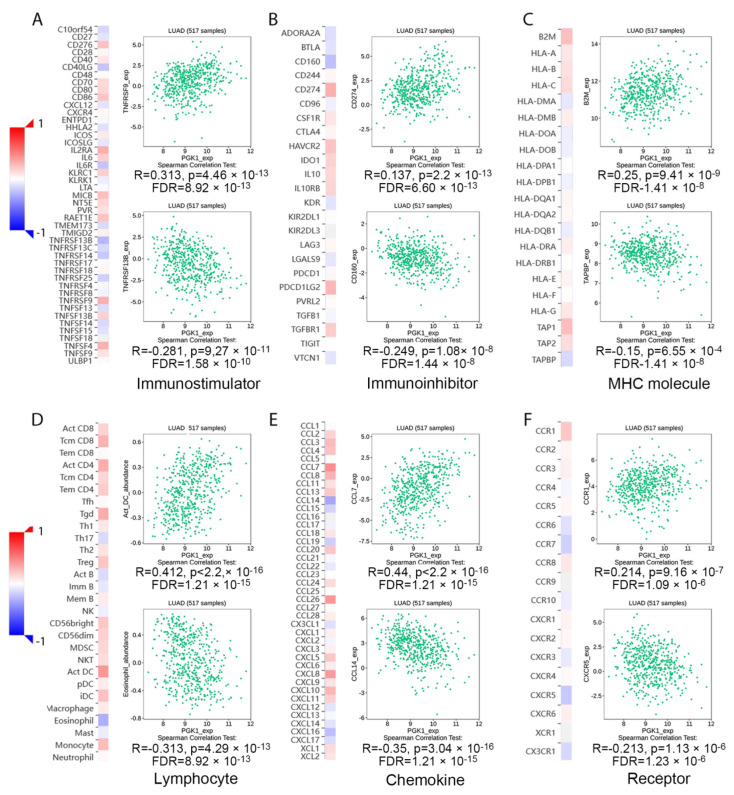
Transcriptional level expression correlation between PGK1 and immunomodulators, chemokines, and lymphocytes. (**A**) Relationship between PGK1 and immunostimulators. (**B**) Relationship between PGK1 and immunoinhibitors. (**C**) Relationship between PGK1 and MHC. (**D**) Relationship between PGK1 and Lymphocytes. (**E**) Relationship between PGK1 and chemokines. (**F**) Relationship between PGK1 and receptors.

**Figure 6 cancers-14-05228-f006:**
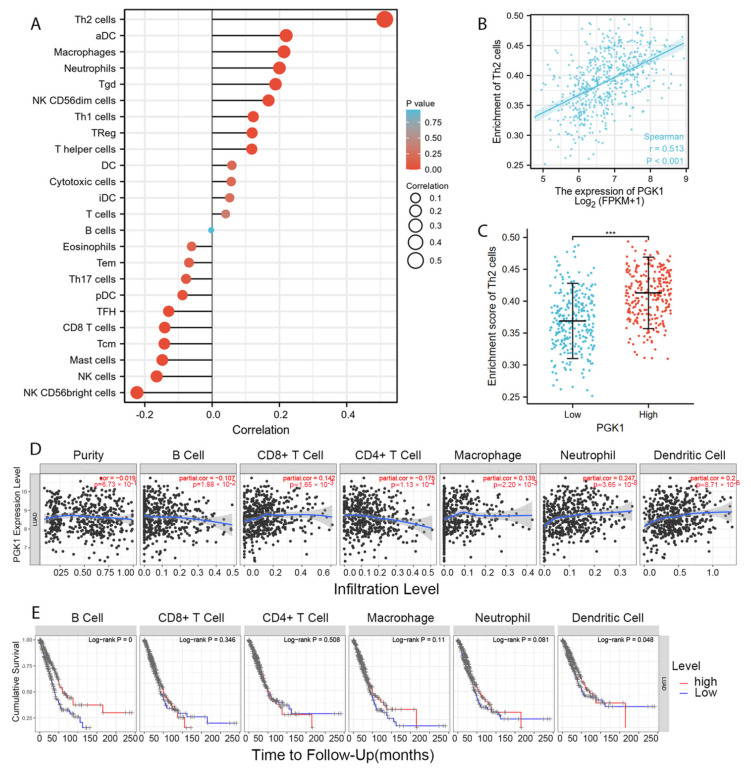
PGK1 expression was linked to immune infiltration in the LUAD microenvironment. (**A**) Forest plots reveal a positive relationship between PGK1 and 13 immune cells, as well as a negative relationship between PGK1 and 11 immune cell types. The size of the dots represents the absolute value of the Spearman’s r. (**B**) Relationship between Th2 cell relative enrichment score and PGK1 expression level (TPM). (**C**) Th2 cell infiltration in the presence of low and high PGK1 expression. (**D**) Relationship between PGK1 expression and immune infiltration in LUAD. (**E**) Kaplan–Meier plots utilized to examine LUAD immune infiltration and overall survival rate. *** *p* < 0.001.

**Figure 7 cancers-14-05228-f007:**
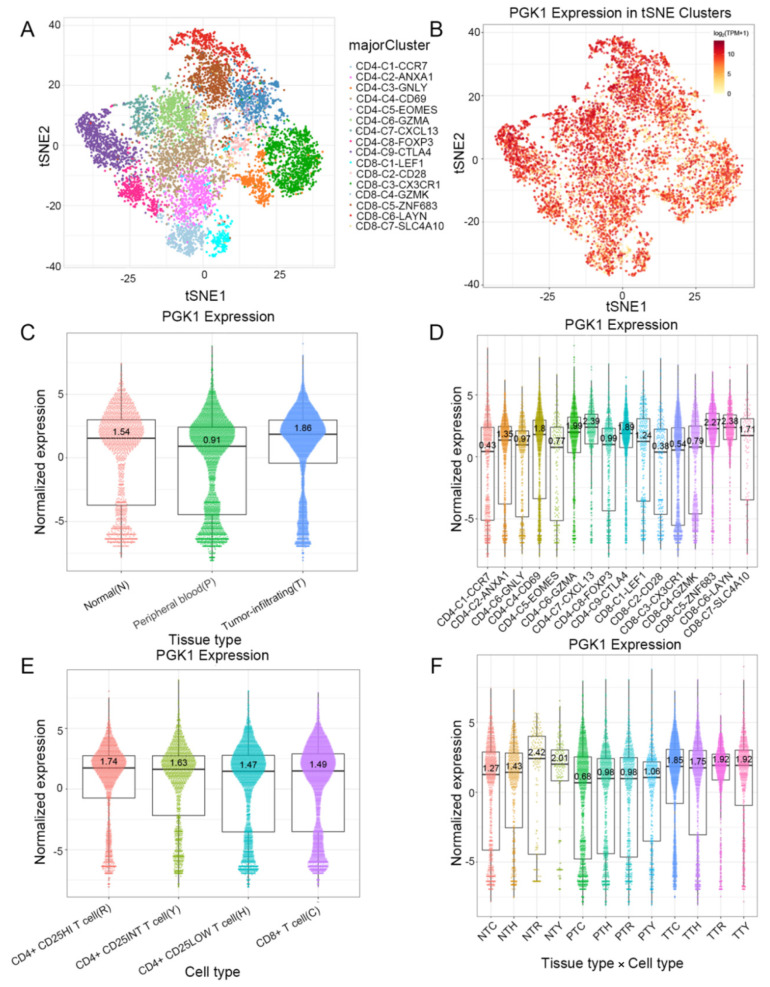
Evaluation of PGK1 expression in global T cells of non-small cell lung cancer in single-cell level. (**A**,**B**) PGK1 expression in tSNE clusters. (**C**) Expression of PGK1 with tissue type and T cell cluster. (**D**) Expression of PGK1 in various T cell clusters. (**E**) Expression of PGK1 by cell type and T cell cluster. (**F**) Expression of PGK1 with respect to tissue type × cell type and T cell cluster.

**Figure 8 cancers-14-05228-f008:**
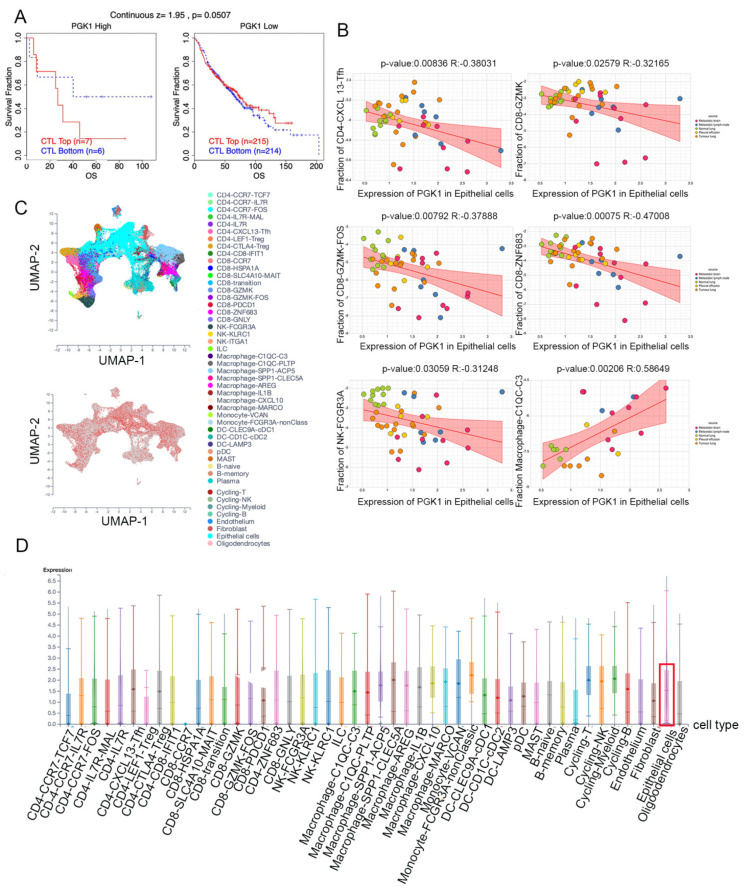
Evaluation of PGK1 expression in tumor cells and immune cells in single-cell level. (**A**) Effect of PGK1 expression on CTL treatment. (**B**) Correlation between PGK1 and immune cell infiltration. (**C**) PGK1 expression in tSNE clusters. (**D**) Expression of PGK1 in tumor cells and immune cells.

**Figure 9 cancers-14-05228-f009:**
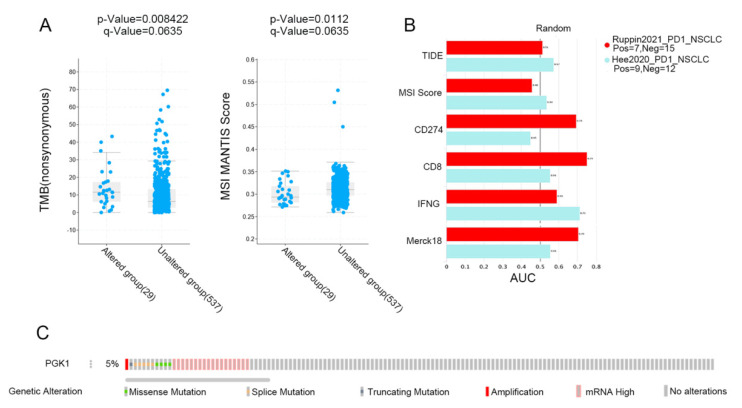
Immunopredictive score of PGK1 in LUAD. (**A**) PGK1 expression related to TMB and MSI in LUAD. (**B**) Common immunopredictive scores of PGK1 in NSCLC including TIDE (Tumor Immune Dysfunction and Exclusion). (**C**) Data of PGK1 altered in LUAD were obtained from the cBioportal database.

**Table 1 cancers-14-05228-t001:** The relationship between PGK1 expression and clinicopathological characteristics in TCGA LUAD samples.

Characteristic	Low Expression of PGK1	High Expression of PGK1	FDR
n	267	268	
T stage, n (%)			0.0345
T1	103 (19.4%)	72 (13.5%)	
T2	130 (24.4%)	159 (29.9%)	
T3	25 (4.7%)	24 (4.5%)	
T4	7 (1.3%)	12 (2.3%)	
N stage, n (%)			0.0456
N0	186 (35.8%)	162 (31.2%)	
N1	39 (7.5%)	56 (10.8%)	
N2	32 (6.2%)	42 (8.1%)	
N3	0 (0%)	2 (0.4%)	
M stage, n (%)			0.0345
M0	179 (46.4%)	182 (47.2%)	
M1	6 (1.6%)	19 (4.9%)	
Pathologic stage, n (%)			0.0345
Stage I	161 (30.6%)	133 (25.2%)	
Stage II	55 (10.4%)	68 (12.9%)	
Stage III	39 (7.4%)	45 (8.5%)	
Stage IV	7 (1.3%)	19 (3.6%)	
Number of packs per year smoked, n (%)			0.0345
<40	110 (29.8%)	78 (21.1%)	
≥40	79 (21.4%)	102 (27.6%)	
Smoker, n (%)			0.3270
No	33 (6.3%)	42 (8.1%)	
Yes	227 (43.6%)	219 (42%)	

## Data Availability

The data sets used in this investigation are available through public repositories. The article/Appendix A contain the names of the repository/repositories.

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
