# Peer review of "Prognosis and Immunological Characteristics of PGK1 in Lung Adenocarcinoma: A Systematic Analysis"

_cancers, 2022, doi:10.3390/cancers14215228_

Round 1

Reviewer 1 Report (Previous Reviewer 1)

The authors have addressed the concerns and suggestions raised in my prior review. 

Author Response

Thank you again for your review

Reviewer 2 Report (New Reviewer)

Major:

1.     The data in Fig.1 are from bulk RNAseq or whole tissue microarray, it's hard to tell the source of PGK1. Given the role of PGK1 in regulating glycolysis, I double the specificity of PGK1expression.

2.     In Fig5, all the results are based on the T cell intrinsic PGK1, but the title of this part is ambiguous.

3.     In Fig6, the authors stressed the correlation between PGK1 and Th2, however, Fig7 showed PGK1 expressed more in the CD4-CXCL13 population, which is Th1 based on the source of the data set.

4.     "Treg represented by CD4+ CD25HI T cell, and the expression of PGK1 was the lowest in 372 Teff represented by CD4+ CD25LOW T cell" I don't think CD25+ CD4 is equal to Treg cells.

5.     According to the recent research, both Treg and Tex cells are mainly dependent on the mitochondria mediated lipid metabolism, given PGK1 is one of the key enzymes controlling glycolysis, which is more required for effector T cell function, I don't buy the idea that the author suggested.

6.     "we also performed single cell sequencing analysis of tumor immune microenvironment in lung adenocarcinoma in GSE131907", please put the tumor specific data in the Fig1, it is very supportive for the point of PGK1 upregulated in the tumor cells.

Minor:

1.     Most labels are too small to read in figures.

2.     The description of Fig8A is missing in the results.

Author Response

This manuscript is a resubmission of an earlier submission. The following is a list of the peer review reports and author responses from that submission.

Round 1

Reviewer 1 Report

This manuscript by Yang et.al., has studied the association of expression level of PGK1 with LUAD prognosis and immunological characteristics using variously available and used bioinformatic tools. Overall, the analysis has been done using appropriate methods. There was no attempt to functionally validate the metabolic-immunological connection. In addition, the assertion from this study for potential “immune-metabolic” combination therapies is premature.  Authors should reword those assertions. Following are the primary concerns of the manuscript in the current form.

1.     This manuscript needs extensive editing.

2.     It is not clear whether multiple hypothesis testing was performed when correlating various clinicopathological characteristics with PGK1 expression.

3.     Tumor microenvironment is “TME”, not “TIME”.

4.     It is not clear from the methods what dataset was used to obtain the protein and phosphorylation data for PGK1. The tool to analyze the data was mentioned- UALCAN. Moreover, what phosphorylation site was assessed, and what is the significance of that?  It seems none of the protein expression differences were significant.  Is that correct?  If so, the discrepancy between protein and RNA expression should be discussed. This could be due to the low sample size of the protein data and also less than optimal protein and phosphorylation analysis. 

5.     What are the potential consequences of the around 2% mutation frequency of PGK1 in LUAD? Are they expected to cause loss of function or gain of function? How does that correlate with expression- higher RNA expression correlates with poorer survival?

6.     Was there CNA (copy number amplification) that correlate with increased expression?

7.     The correlation with immunomodulators is relatively low although significant p-values are shown.  Again- was there multiple hypothesis testing incorporated in this analysis? 

8.     From scRNA seq data- can the authors say whether PGK1 is expressed more in tumor cells vs immune cells or other cells in TME? 

9.     Authors should discuss the paper in the context of published literature on PGK1 and cancer prognosis, drug resistance etc. (e.g. He et al., Am J of Cancer Res 2019; 9(11): 2280-2300; C hang et al., Cell Death Discovery (2021)7:135; etc.)   

Author Response

Thank you very much for your question, please check the attachment.

Reviewer 2 Report

1. The biggest concern is this manuscript consisting solely of bioinformatics or computational analysis databases which are not accompanied by validation

2. Authors have published a paper with high similarity to this one (e.g. Systematic Analysis Uncovers Associations of PGK1 with Prognosis and Immunological Characteristics in Breast Cancer)

3. The role of PGK1 in immune regulations has been widely reported. For example, macrophages regulate tumor cell aerobic glycolysis and progression via PGK1.

The findings therefrom are not sufficiently in depth to be novel or impactful.

Author Response

(The authors gave the same response as above.)

Round 2

Reviewer 2 Report

Almost all genes play different roles in the context of physiology and pathology. Similar approaches have been applied for analyzing PGK1 in previous study. This is just another cancer type. The author cannot fundamentally address my concerns about novelty.